# Evaluation of Adverse Reactions to Influenza Vaccination: A Prospective Cohort Study

**DOI:** 10.3390/vaccines10101664

**Published:** 2022-10-06

**Authors:** Ayako Kumabe, Tsuneaki Kenzaka, Shinsuke Yahata, Ken Goda, Masanobu Okayama

**Affiliations:** 1Division of Community Medicine and Career Development, Kobe University Graduate School of Medicine, Kobe 650-0017, Japan; 2Department of Internal Medicine, Toyooka Public Hospital, Toyooka 668-8501, Japan; 3Department of Internal Medicine, Hyogo Prefectural Tamba Medical Center, Tamba 669-3495, Japan; 4Department of Internal Medicine, Shiso Municipal Hospital, Shiso 671-2576, Japan; 5Department of General Internal Medicine, Hyogo Prefectural Harima-Himeji General Medical Center, Himeji 670-8560, Japan; 6Division of Community Medicine and Medical Education, Kobe University Graduate School of Medicine, Kobe 650-0017, Japan

**Keywords:** influenza vaccine, adverse reaction, sex, age, influenza morbidity, influenza vaccination history

## Abstract

This study aimed to investigate the influence of sex, age, and quadrivalent vaccination history on adverse reactions (ARs) to influenza vaccines and the relationship between the occurrence of ARs and the risk of influenza infection. Study participants were employees of three hospitals in the Hyogo Prefecture, Japan, who received the influenza vaccine in 2019. Data were collected using questionnaires. The main factors were age, sex, and history of influenza vaccination as a control. The primary outcomes were the incidence of local and systemic ARs attributable to the vaccine and positive influenza cases among the participants during the influenza season. Logistic regression was used to calculate the odds ratio (OR) and 95% confidence interval (CI). Among the 1493 participants, 80% experienced either local or systemic ARs. ARs were less common among men than among women (OR: 0.28, 95% CI: 0.21–0.37) and less common among those aged ≥60 years (OR: 0.48, 95% CI: 0.26–0.89). ARs were significantly more likely to occur in those with a history of influenza vaccination (OR: 1.96, 95% CI: 1.15–3.33). Those who had ARs, notably localized ones, were significantly more likely to incur influenza infection. Individuals who report ARs to influenza vaccination should strictly adopt non-pharmaceutical preventive measures in the hospital, community settings, and at home.

## 1. Background

Although the number of influenza cases has been progressively decreasing since 2020, in part due to the health and safety precautions adopted after the outbreak of the coronavirus disease [1,2], the incidence of influenza usually increases during the winter season. In Japan, which has a total population of 124.84 million, 12 million people (9.5% of the population) were estimated to have been affected during the 2018–2019 season and 7.28 million (5.7% of the population) during the 2019–2020 season [3]. In the 2020–2021 season, the cumulative number of influenza cases was 14,000 [2], with 956 deaths [4], a significant decrease compared to that in previous years owing to the impact of the novel coronavirus epidemic and its countermeasures.

A surveillance conducted by the Japanese Respiratory Society showed that the median age in influenza cases requiring hospitalization in five seasons during 2015–2019 was 78 years, and in all seasons, patients were in their 80s and/or 70s. In-hospital deaths occurred in 4.8% of cases, the median age at death was 82 years, and pneumonia was the reason for hospitalization in >60% of the cases [5].

The elderly have higher rates of morbidity and mortality from influenza infections than younger adults, and influenza-related deaths have been reported in many elderly cases [5,6]. Therefore, the prevention of infection by close surveillance to monitor infection trends is and will continue to be important for this disease.

Influenza vaccines are effective for preventing infection in healthy adults; however, their efficacy is reported to be lower among the elderly [7]. Despite this, influenza vaccine is reported to be effective in preventing severity and hospitalization [8], particularly among patients at risk of severe complications, such as the elderly, and those with comorbidities at any age.

During the 2013–2014 season, the World Health Organization recommended the quadrivalent vaccine consisting of four vaccine strains, including two strains each of type A and B influenza viruses, instead of the trivalent vaccine consisting of two strains of type A and one strain of type B influenza virus [9]. In Japan, the quadrivalent vaccine has been used since the 2015–2016 season [10]. Although the literature on adverse reactions post influenza vaccination is limited, adverse reactions are more likely to occur in women than in men with both the quadrivalent and trivalent vaccines [11] and less likely to occur in older people [10].

However, the influence of vaccination history on adverse reactions to influenza vaccines and the relationship between the occurrence of adverse reactions and the risk of influenza are not yet established.

The purpose of this study was to investigate the influence of sex, age, and quadrivalent vaccination history on adverse reactions to the influenza vaccine and the relationship between the occurrence of adverse reactions and risk of influenza infection.

## 2. Materials and Methods

### 2.1. Study Design

This prospective cohort study was approved by the Ethics Committee of the Hyogo Prefectural Tamba Medical Center (Approval No. Tan-I No. 1166). The questionnaire distributed to the participants stated that the collected data are to be used for research purposes, and written informed consent was obtained for the publication of the study results. 

### 2.2. Participants and Setting

Employees of three hospitals in Hyogo Prefecture (Hyogo Prefectural Tamba Medical Center, Public Toyooka Hospital, and Shiso Municipal Hospital) who received the 2019 influenza vaccine were eligible. The influenza vaccine administered in the 2019–2020 season included the following four subtypes: A/Brisbane/02/2018(H1N1)pdm09-like virus, A/Kansas/14/2017 (H3N2)-like virus, B/Colorado/06/2017-like virus (B/Victoria/2/87 lineage), B/Phuket/3073/2013-like virus (B/Yamagata/16/88 lineage).

### 2.3. Questionnaire Items

We distributed two types of questionnaires. The first questionnaire collected participants’ demographic data and identified participants who had received the vaccination. The second questionnaire collected data regarding adverse reactions to the vaccine 10 days after immunization. 

Demographic data included the following items: age, sex, pregnancy (women only), temperature at the time of vaccination, presence or absence of any illness at the time of vaccination, history of influenza vaccination, and any history of food and/or drug allergies.

Local adverse reactions included redness, swelling, induration, pain, heat, itchiness, and heaviness/tingling at the vaccination site. In contrast, systemic adverse reactions included fever or chills, headache, fatigue, nasal discharge, cough, nausea, diarrhea, difficulty moving upper extremities, and numbness.

### 2.4. Identification of Influenza Cases

Preliminary questionnaires were numbered and were administered by the infection control department of each hospital, independent of the investigators. The infection control department of each hospital kept track of the employees who received the 2019 influenza vaccination and were infected with influenza during the 2019–2020 season. These data were used to match the affected personnel to the numbers on the questionnaire to determine if they contracted influenza during the season.

With this method, information on the demographic characteristics of the participants, the presence of adverse reactions, and incidence of influenza was blinded from the researchers, thus protecting their personal information.

### 2.5. Data Analysis 

The demographic characteristics of the participants were tabulated. Any missing values were excluded, and subsequent calculations were also removed from the analysis.

### 2.6. Primary Factors

Participants with no adverse reactions were stratified according to age (<40, 40–59, >60 years), sex, and history of influenza vaccination as a control.

### 2.7. Main Outcome Indicators

Influenza cases during the season: No/Yes

Adverse reactions after vaccination

Adverse reactions: None/any of the local and systemic adverse reactions

Local adverse reactions: None/any of the local adverse reactions

Systemic adverse reactions: None/any of the systemic adverse reactions

### 2.8. Statistical Analysis 

The number of adverse reactions for each factor was tabulated. A logistic regression analysis was used to calculate the odds ratio (OR) and 95% confidence interval (CI) for each of the primary factors against each primary outcome measure. The analysis was performed using the following three models.

Model 1: Crude analysis. 

Model 2: Adjusted for sex and age.

Model 3: Model 2 plus adjustment for influenza vaccination history.

Stata MP version 15 (StataCorp, College Station, TX, USA) was used for the analyses.

## 3. Results

At Hyogo Prefectural Tamba Medical Center, out of 717 respondents, 657 completed the questionnaire (response rate: 91.63%) and were immunized against influenza. At Public Toyooka Hosptial, 622 out of 999 (response rate: 62.26%) respondents completed the questionnaire and were immunized against influenza. At Shiso Municipal Hospital, 245 out of 314 (response rate: 78.03%) respondents completed the questionnaire and were immunized against influenza. Overall, out of 2030 individuals who were immunized, 1524 responded to the questionnaires (recovery rate, 75.07%). Of the 1524 participants, 31 were excluded due to undisclosed age. Finally, 1493 participants were included in the study. The basic attributes and occurrence of local and/or systemic adverse reactions in the included participants are summarized in Table 1. The mean age was 42.4 years, with a standard deviation of ± 12.8 years, and 76.29% of the participants were women. All participants were aged between 20 and 70 years; 622 (41.66%) were aged <40 years, 818 (54.79%) were aged 40–64 years, and 53 (3.55%) were aged ≥65 years. 

The majority (95.08%) of participants had been previously vaccinated against influenza. Among the participants, 64 (4.29%) contracted influenza, and 1182 (79.17%), 261 (17.48%), and 1193 (79.91%) reported local adverse reactions, systemic adverse reactions, and both local and systemic adverse reactions, respectively. Although data on individual influenza subtypes in the 64 participants who contracted influenza are not available, the influenza types and subtypes that were prevalent in Japan during the 2019–2020 season were A/H1N1pdm09 (85%), A/H3N2 (2%), and B (12%; Victoria strain 98.7%, Yamagata strain 0.6%, and unknown strain 0.7%). In addition, influenza vaccines administered in 2019 contained the following strains: A/Brisbane/02/2018(H1N1)pdm09-like virus, A/Kansas/14/2017 (H3N2)-like virus, B/Colorado/06/2017-like virus (B/Victoria/2/87 lineage), and B/Phuket/3073/2013-like virus (B/Yamagata/16/88 lineage) [3]. Therefore, it can be considered that the strains in the vaccine and the affected participants were almost identical.

Table 2 shows the factors associated with any adverse reaction (either local or systemic adverse reaction) on the three models. Adverse reactions were significantly less likely to occur in men (adjusted OR, 0.28 [95% CI, 0.21–0.37]). Adverse reactions tended to decrease with age and were significantly less likely to occur in persons aged ≥65 years (adjusted OR, 0.48 [95% CI, 0.26–0.89]). Participants with a history of influenza vaccination were significantly more likely to have an adverse reaction (adjusted OR, 1.96 [95% CI, 1.15–3.33]).

Table 3 illustrates the factors associated with local adverse reactions in the three models. Adverse reactions were significantly less likely to occur in men (adjusted OR, 0.27 [95% CI, 0.20–0.35]). Adverse reactions tended to decrease with age and were significantly less likely to occur in persons aged ≥65 years (adjusted OR, 0.48 [95% CI, 0.26–0.89]). Participants with a history of influenza vaccination were significantly more likely to have a local adverse reaction (adjusted OR, 1.96 [95% CI, 1.15–3.33]).

Table 4 summarizes the factors associated with systemic adverse reactions in the three models. Adverse reactions were significantly less likely to occur in men (adjusted OR, 0.55 [95% CI, 0.38–0.79]). The likelihood of systemic adverse reactions was not influenced by age or history of influenza vaccination.

Table 5 shows the impact of each adverse reaction on influenza morbidity. Experience of any adverse reaction, especially local adverse reactions, had a significantly higher adjusted OR for influenza incidence. In particular, local adverse reactions such as redness (adjusted OR, 2.92 [95% CI, 1.51–5.65]), swelling (adjusted OR, 2.00 [95% CI, 1.09–3.65]), heat sensation (adjusted OR, 1.75 [95% CI, 1.02–3.02]), and itching (adjusted OR, 1.90 [95% CI, 1.10–3.27]) had significantly higher adjusted ORs for influenza incidence. Among systemic adverse reactions, diarrhea was associated with a significantly higher adjusted OR for influenza incidence.

## 4. Discussion

This study showed a trend for fewer adverse reactions with increased age. Particularly, participants aged >65 years experienced significantly fewer adverse reactions. Moreover, men had significantly fewer adverse reactions than women. Adverse reactions were significantly more likely to occur in participants who had received an influenza vaccination in the past. Influenza infection was significantly more common in participants reporting any adverse reactions (especially local adverse reactions). The results are further discussed in detail below.

### 4.1. Incidence of Influenza and Adverse Reactions 

In a Japanese study of 3275 cases at 97 facilities, the incidence of adverse reactions was 86.8%. The incidence of adverse reactions was higher in women than in men (91.8% vs. 82.4%), with a tendency for fewer adverse reactions in older age groups (87.3% and 71.3% in participants aged 15–64 years and ≥65 years, respectively) [10,11].

In studies conducted in Australia and the Philippines on healthy adults, the incidence of both local and systemic adverse reactions was reported to be approximately 50% [12], while in a study conducted in Japan on medical professionals, the incidence of local and systemic reactions was reported to be 73.9% and 15.8%, respectively [10]. The incidence of adverse reactions in this study was 79.17% and 17.48% for local and systemic reactions, respectively, which is similar to that reported in previous studies.

In this study, 64 of the 1524 participants (4.29%) contracted influenza. The incidence of influenza in Japan during the same observation period, was 5.7% [3]. Considering that the participants in this study were healthcare workers, who are more knowledgeable regarding infection control measures than the general public, we believe that this incidence rate is reasonable.

In the 2019–2020 season, the overall incidence of influenza in Japan was 16.1% in the 20–39 years age group (6.4% in the 20–29 years age group and 9.7% in the 30–39 years age group), 19.5% in the 40–59 years age group (12.2% in the 40–49 years age group and 7.3% in the 50–59 years age group), and 4.8% in the 60–69 years age group [3]. The modality of collection of information is active. In our study, no patient aged between 65 and 70 years contracted influenza, which was the lowest incidence for this age group in Japan. Therefore, our results are not inconsistent with those for Japan as a whole.

Based on the above, we believe that the incidence of influenza and adverse reactions in this study, which were used as the primary outcome measures, were reasonable.

### 4.2. Age and Adverse Reactions

Many factors are involved in immunity; however, it is well-established that as age increases, the immune system declines, which translates to a decline in both innate and acquired immunity to infection [13]. Therefore, resistance to infection declines, making prevention important with annual influenza vaccination recommended for the elderly. Vaccine efficacy in preventing the onset of influenza is 70–90% in healthy adults, but only 17–53% in the elderly [7].

In immunological aging, regulatory T cells are inappropriately suppressed, resulting in autoantibody production that reduces immune function [14]. Moreover, CD28+ expression is said to be directly related to influenza vaccine response [14]. Because CD28+ expression of T cells is lower among the elderly, vaccine adverse reactions are less likely to appear, and vaccine efficacy decreases with increasing age [14].

The influenza vaccine is less effective in preventing disease in the elderly than in healthy adults [8]. However, it is effective not only in preventing the onset of influenza but also in preventing hospitalization (prevention of severe illness) [7], which may play an important role in the elderly, who are frequently hospitalized.

In the elderly, vaccination is expected to prevent the onset and severity of disease, whereas in healthy adults, vaccination is effective for preventing the onset of disease in the vaccinee and transmission of infection to the elderly or immunocompromised individuals [8]. In particular, healthcare workers have many opportunities to come into contact with patients with underlying diseases, and vaccination is meaningful not only for preventing their own infection, but also from the perspective of protecting patients. 

### 4.3. Sex and Adverse Reactions

In terms of sex, women have higher immunity than men [15], and past studies on adverse reactions to influenza vaccines have reported higher rates of adverse reactions among women than among men [15].

Due to the differences in the effects of sex hormones on immune cells, genetic factors, and microflora, higher incidence of adverse reactions to vaccines and higher antibody production have been reported in women than in men [15]. Similarly, in the present study, women had significantly more adverse reactions.

### 4.4. History of Influenza Vaccination and Adverse Reactions

Participants with a history of influenza vaccination had a significantly higher incidence of local adverse reactions than those with no history of influenza vaccination. This may be due to the repeated exposure of the immune system to the antigen, which activates memory T cells, leading to a faster and greater response [13] than with the initial exposure.

### 4.5. Influenza Morbidity and Adverse Reactions

Regarding the relationship between adverse reactions and influenza morbidity, our findings suggest that the risk of influenza morbidity may be higher when some adverse reactions (especially local adverse reactions) are observed. In this study, the adjusted ORs suggested a tolerable incidence of influenza morbidity. With regard to the severe acute respiratory syndrome coronavirus 2, greater adverse reactions experienced post-vaccination translates to a higher antibody production [16], which reduces the susceptibility to coronavirus disease 2019. In this study, more adverse reactions resulted in a higher incidence of influenza. The mechanism for this is unknown and has not been reported in the existing literature. In the current study, a history of influenza vaccination significantly increased the incidence of any adverse reaction (either local or systemic) and local adverse reactions. However, 95.08% of participants had a history of previous influenza vaccination. This may explain why the occurrence of adverse reactions did not necessarily reduce the incidence of influenza.

### 4.6. Limitations

In this study, participants self-reported the occurrence of adverse reactions. Since the participants were healthcare professionals, the possibility of subjectivity and selection bias cannot be ruled out, although a certain degree of reliability was maintained. However, caution should be exercised when generalizing the results.

We did not investigate whether past influenza vaccinations were trivalent or quadrivalent. However, given that participants received the influenza vaccine from the hospital and at an almost yearly interval, as hospital personnel, we believe that they have been vaccinated almost exclusively with quadrivalent vaccinations since the 2015–2016 season.

The participants in this study were hospital employees in good health aged between 20 and 70 years. Therefore, people in their late 70s and aged >80 years, who have high rates of serious illness and mortality, were not included in the study. Hence, care should be taken when generalizing the results to this age group.

## 5. Conclusions

In this study, adverse reactions were less likely to occur with increasing age in healthcare professionals, especially in participants aged >65 years. Moreover, men were significantly less likely to have an adverse reaction than women.

Furthermore, participants with a history of influenza vaccination were more likely to have adverse reactions. In terms of the relationship between adverse reactions and influenza morbidity, participants who had adverse reactions, especially local adverse reactions, were significantly more likely to contract influenza.

Individuals reporting adverse reactions to influenza vaccination should strictly adhere to non-pharmaceutical preventive measures in hospital and community settings as well as at home.

## Figures and Tables

**Table 1 vaccines-10-01664-t001:** Participants’ background characteristics and symptoms in the 10 days after vaccination according to age.

Background Characteristics	Overall		<40 Years		40–64 Years		≥65 Years	
	n = 1493		n = 622 (41.66%)		n = 818 (54.79%)		n = 53 (3.55%)	
	n	%	N	%	n	%	N	%
Sex								
Female	1139	76.34	465	74.76	647	79.19	27	50.94
Male	353	23.66	157	25.24	170	20.81	26	49.06
Unknown	1		0		1		0	
Age (years: mean, SD)	42.4	12.80	29.82	5.83	50.24	6.60	69.02	2.78
Pregnant								
No	1058	97.78	421	95.25	611	99.51	26	100.00
Yes	24	2.22	21	4.75	3	0.49	0	0.00
Unknown	57		23		33		1	
Body temperature at the time of vaccination (°C: mean, SD)	36.39	0.36	36.48	0.33	36.34	0.36	36.2	0.35
Unknown	86							
Physical condition at the time of vaccination								
Good	1435	98.15	591	97.20	791	98.75	53	100.00
Poor	27	1.85	17	2.80	10	1.25	0	0.00
Unknown	31		14		17		0	
History of influenza vaccination								
No	73	4.92	36	5.83	35	4.29	2	3.77
Yes	1412	95.08	581	94.17	780	95.71	51	96.23
Unknown	8		5		3		0	
Food/drug allergies								
No	1301	88.02	545	88.05	708	87.84	48	90.57
Yes	177	11.98	74	11.95	98	12.16	5	9.43
Unknown	15		3		12		0	
Contracted influenza								
No	1429	95.71	589	94.69	787	96.21	53	100.00
Yes	64	4.29	33	5.31	31	3.79	0	0.00
Unknown	0		0		0		0	
Redness at the injection site								
No	567	38.10	235	37.90	301	36.93	31	58.49
Yes	921	61.90	385	62.10	514	63.07	22	41.51
Unknown	5		2		3		0	
Swelling at the injection site								
No	563	37.96	220	35.60	308	37.93	35	66.04
Yes	920	62.04	398	64.40	504	62.07	18	33.96
Unknown	10		4		6		0	
Induration at the injection site								
No	1059	71.55	447	72.33	571	70.49	41	78.85
Yes	421	28.45	171	27.67	239	29.51	11	21.15
Unknown	13		4		8		1	
Pain at the injection site								
No	791	53.34	305	49.35	450	55.35	36	69.23
Yes	692	46.66	313	50.65	363	44.65	16	30.77
Unknown	10		4		5		1	
Heat sensation at the injection site								
No	731	49.09	300	48.39	395	48.35	36	69.23
Yes	758	50.91	320	51.61	422	51.65	16	30.77
Unknown	4		2		1		1	
Itching at the injection site								
No	800	53.76	334	53.70	427	52.39	39	76.47
Yes	688	46.24	288	46.30	388	47.61	12	23.53
Unknown	5		0		3		2	
Heaviness/lassitude at the injection site								
No	1208	81.18	514	82.90	648	79.51	46	86.79
Yes	280	18.82	106	17.10	167	20.49	7	13.21
Unknown	5		2		3		0	
Other localized symptoms								
No	1454	98.18	614	99.19	789	97.29	51	100.00
Yes	27	1.82	5	0.81	22	2.71	0	0.00
Unknown	12		3		7		2	
Some types of localized symptoms								
No	311	20.83	129	20.74	161	19.68	21	39.62
Yes	1182	79.17	493	79.26	657	80.32	32	60.38
Unknown	0		0		0		0	
Fever								
No	1467	98.26	609	97.91	806	98.53	52	98.11
Yes	26	1.74	13	2.09	12	1.47	1	1.89
Unknown	0		0		0		0	
Chills								
No	1471	98.59	613	98.71	806	98.53	52	98.11
Yes	21	1.41	8	1.29	12	1.47	1	1.89
Unknown	1		1		0		0	
Headache								
No	1429	95.71	592	95.18	786	96.09	51	96.23
Yes	64	4.29	30	4.82	32	3.91	2	3.77
Unknown	0		0		0		0	
Fatigue								
No	1382	92.57	578	92.93	753	92.05	51	96.23
Yes	111	7.43	44	7.07	65	7.95	2	3.77
Unknown	0		0		0		0	
Nasal discharge								
No	1499	97.15	606	97.43	791	96.70	52	98.11
Yes	44	2.85	16	2.57	27	3.30	1	1.89
Unknown	0		0		0		0	
Cough								
No	1465	98.12	613	98.55	800	97.80	52	98.11
Yes	28	1.88	9	1.45	18	2.20	1	1.89
Unknown	0		0		0		0	
Nausea								
No	1480	99.13	615	98.87	812	99.27	53	100.00
Yes	13	0.87	7	1.13	6	0.73	0	0.00
Unknown	0		0		0		0	
Diarrhea								
No	1475	98.86	615	98.87	807	98.78	53	100.00
Yes	17	1.14	7	1.13	10	1.22	0	0.00
Unknown	1		0		1		0	
Difficulty moving the upper limbs								
No	1397	93.70	586	94.21	759	93.01	52	98.11
Yes	94	6.30	36	5.79	57	6.99	1	1.89
Unknown	2		0		2		1	
Numbness								
No	1472	98.66	616	99.04	803	98.29	53	100.00
Yes	20	1.34	6	0.96	14	1.71	0	0.00
Unknown	1		0		1		0	
Other systemic symptoms								
No	1461	98.65	608	98.54	800	98.64	53	100.00
Yes	20	1.35	9	1.46	11	1.36	0	0.00
Unknown	12		5		7		0	
Some types of systemic symptoms								
No	1232	82.52	522	83.92	662	80.93	48	90.57
Yes	261	17.48	100	16.08	156	19.07	5	9.43
Unknown	0		0		0		0	
Some types of systemic or localized symptoms								
No	300	20.09	124	19.94	155	18.95	21	39.62
Yes	1193	79.91	498	80.06	663	81.05	32	60.38
Unknown	0		0		0		0	

SD, standard deviation.

**Table 2 vaccines-10-01664-t002:** Factors associated with any adverse reaction (either local or systemic).

	OR (95% CI)		
	Model 1	Model 2	Model 3
**Sex**			
Female	Reference	Reference	Reference
Male	0.27 (0.20–0.35)	0.28 (0.21–0.36)	0.28 (0.21–0.37)
**Age**			
<40 years	Reference	Reference	Reference
40–64 years	1.07 (0.82–1.39)	1.00 (0.76–1.31)	0.98 (0.74–1.29)
≥65 years	0.38 (0.21–0.68)	0.49 (0.27–0.91)	0.48 (0.26–0.89)
**History of influenza vaccination**			
No	Reference	Reference	Reference
Yes	2.03 (1.22–3.36)	-	1.96 (1.15–3.33)

Model 1. Rough analysis; Model 2. Adjusted for sex and age; Model 3. Adjusted for sex, age, and history of influenza vaccination. OR, odds ratio; CI, confidence interval.

**Table 3 vaccines-10-01664-t003:** Factors associated with local adverse reactions.

	OR (95% CI)		
	Model 1	Model 2	Model 3
**Sex**			
Female	Reference	Reference	Reference
Male	0.26 (0.20–0.34)	0.27 (0.2–0.35)	0.27 (0.20–0.35)
**Age**			
<40 years	Reference	Reference	Reference
40–64 years	1.07 (0.82–1.39)	1.00 (0.76–1.31)	0.98 (0.74–1.29)
≥65 years	0.38 (0.21–0.68)	0.49 (0.27–0.91)	0.48 (0.26–0.89)
**History of influenza vaccination**			
No	Reference	Reference	Reference
Yes	2.03 (1.22–3.36)	-	1.96 (1.15–3.33)

Model 1. Rough analysis; Model 2. Adjusted for sex and age; Model 3. Adjusted for sex, age, and history of influenza vaccination. OR, odds ratio; CI, confidence interval.

**Table 4 vaccines-10-01664-t004:** Factors associated with systemic adverse reactions.

	OR (95% CI)		
	Model 1	Model 2	Model 3
**Sex**			
Female	Reference	Reference	Reference
Male	0.53 (0.37–0.76)	0.55 (0.38–0.79)	0.55 (0.38–0.79)
**Age**			
<40 years	Reference	Reference	Reference
40–64 years	1.23 (0.93–1.62)	1.20 (0.91–1.59)	1.20 (0.91–1.58)
≥65 years	0.54 (0.21–1.40)	0.62 (0.24–1.60)	0.61 (0.24–1.59)
**History of influenza vaccination**			
No	Reference	Reference	Reference
Yes	1.35 (0.68–2.67)	-	1.30 (0.65–2.57)

Model 1. Rough analysis; Model 2. Adjusted for sex and age; Model 3. Adjusted for sex, age, and history of influenza vaccination. OR, odds ratio; CI, confidence interval.

**Table 5 vaccines-10-01664-t005:** Factors associated with influenza incidence.

	OR (95% CI)		
	Model 1	Model 2	Model 3
**Local site adverse reaction**			
Redness	2.50 (1.35–4.64)	2.83 (1.47–5.44)	2.92 (1.51–5.65)
Swelling	1.88 (1.06–3.35)	1.94 (1.06–3.53)	2.00 (1.09–3.65)
Induration	1.65 (0.99–2.76)	1.68 (1.00–2.84)	1.67 (0.99–2.12)
Pain	0.88 (0.53–1.47)	0.85 (0.51–1.41)	0.87 (0.52–1.44)
Heat sensation	1.64 (0.98–2.75)	1.71 (1.00–2.95)	1.75 (1.02–3.02)
Itching	1.74 (1.05–2.90)	1.88 (1.09–3.24)	1.90 (1.10–3.27)
Heaviness/lassitude	0.91 (0.47–1.77)	0.94 (0.48–1.83)	0.95 (0.49–1.86)
Other localized symptoms	0.86 (0.12–6.47)	0.94 (0.13–7.10)	0.96 (0.13–7.27)
Some types of localized symptoms	2.20 (0.99–4.87)	2.31 (1.02–5.22)	2.41 (1.06–5.48)
**Systemic adverse reaction**			
Fever	-	-	-
Chills	-	-	-
Headache	0.71 (0.17–2.97)	0.71 (0.17–2.99)	0.70 (0.17–2.96)
Fatigue	1.06 (0.42–2.69)6	1.07 (0.42–2.73)	1.07 (0.42–2.73)
Nasal discharge	1.66 (0.50–5.53)	1.79 (0.54–5.96)	1.83 (0.55–6.10)
Cough	1.74 (0.40–7.50)	1.85 (0.43–8.02)	1.83 (0.42–7.97)
Nausea	1.87 (0.24–14.64)	1.71 (0.22–13.55)	1.64 (0.20–13.06)
Diarrhea	4.97 (1.39–17.74)	4.70 (1.31–16.93)	4.40 (1.21–16.08)
Difficulty moving the upper limbs	0.99 (0.35–2.79)	1.00 (0.36–2.82)	1.05 (0.37–2.96)
Numbness	1.12 (0.15–8.46)	1.18 (0.16–8.97)	1.23 (0.16–9.34)
Other systemic symptoms	-	-	-
Some types of systemic symptoms	0.97 (0.50–1.89)	1.00 (0.51–1.95)	1.01 (0.52–1.97)
**Some types of systemic or localized symptoms**	2.49 (1.07–5.83)	2.62 (1.10–6.25)	2.75 (1.15–6.60)

Model 1. Rough analysis; Model 2. Adjusted for sex and age; Model 3. Adjusted for sex, age, and history of influenza vaccination. OR, odds ratio; CI, confidence interval.

## Data Availability

The data sets used and/or analyzed during the present study are available from the first author upon reasonable request.

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
