# Peer review of "Evaluation of Adverse Reactions to Influenza Vaccination: A Prospective Cohort Study"

_vaccines, 2022, doi:10.3390/vaccines10101664_

Round 1

Reviewer 1 Report

          The paper Evaluation of adverse reactions to Influenza Vaccination: A Prospective cohort study has been reviewed.  First of all, thank you for the opportunity to review this paper. The most common adverse effects of influenza vaccinations administered by injection in both adults and children are local symptoms. They include pain, swelling, redness and hotness at the injection site. Fever is more common in children under the age of 2 with skin reactions especially in connection with fever. Rare cases present joint inflammations, transient reduction in the platelet count or neurological adverse effects have also been reported. As to most common side effects of the nasal spray vaccine, a blocked or very runny nose is about the only AR observed. 

All of these have been widely described, yet there is less information as to how repeated immunization relates to ARs or how ARs relate to higher risk of attaining Influenza infection.

This survey reveals information as to the factors related to ARs in influenza vaccination of HCWs in several hospital settings in Japan.  

Here are my considerations for corrections and changes.

Note: The naming of the vaccine should be the same throughout the entire paper, either tetravalent or quadrivalent vaccine , either one is correct but stick to one of them.

Abstract

Conclusion

Those who had ARs, notably localized ones, were  significantly more likely to incur influenza infection. Individuals who have had ARs to influenza  vaccination should be more cautious about seasonal influenza illness

I would be more concise as to being more cautious, for example: strictly adhering to Non-pharmaceutical interventions in the hospital setting as well as in the community setting and household.

Background

Lines 51-52

Therefore, prevention of infection is and will continue to be important for this disease, which requires careful attention to determine infection trends.

Change: careful attention to close surveillance to monitor infection trends

Lines 53-56

Influenza vaccines are effective in preventing infection in healthy adults; however,   effectiveness is reported to be lower among the elderly than among the youth [8]. Despite   this, the influenza vaccine has been reported to be effective in preventing severity and   hospitalization in addition to preventing infection [9] and may be important for the elderly.  

Rephrase:

Influenza vaccines are effective in preventing infection in healthy adults; however,   effectiveness is reported to be lower among the elderly [8]. Despite this, influenza vaccine has been reported to be effective in preventing severity and   hospitalization [9] especially among population at risk of severe complications such as  the elderly and those with comorbidities at any age.

Lines 62-64:

There is limited literature on adverse reactions post influenza vaccination, of which, adverse reactions have been reported to more likely occur in women than in men with both the quadrivalent and trivalent vaccines [12] and that adverse reactions are less likely to occur in older people [11].

The structure of the sentences is awkward. Change to:

There is limited literature on adverse reactions post influenza vaccination, most of which refers to adverse reactions being more likely to occur in women than in men with both the quadrivalent and trivalent vaccines [12] and that adverse reactions are less likely to occur in older people [11].

2.Materials and Methods

An explanation as to what is considered history of influenza vaccination is needed. Did you considere any previous immunization or the last season, last 2-3-4-5 seasons?

The questionnaire distributed to the participants stated that it is to be used for  research purposes, and informed consent was obtained in writing for the publication of  76 the study results. written informed consent was obtained for the publication of the study results

 Setting Line 52

The study questionnaire and immunization questionnaire were distributed to the  participants ? There are two questionnaires ? what is the difference between them ?

Conclusions : Same comment as in abstract conclusions

Author Response

Reviewer 1’s Comments and Suggestions for Authors:

  The paper Evaluation of adverse reactions to Influenza Vaccination: A Prospective cohort study has been reviewed.  First of all, thank you for the opportunity to review this paper. The most common adverse effects of influenza vaccinations administered by injection in both adults and children are local symptoms. They include pain, swelling, redness and hotness at the injection site. Fever is more common in children under the age of 2 with skin reactions especially in connection with fever. Rare cases present joint inflammations, transient reduction in the platelet count or neurological adverse effects have also been reported. As to most common side effects of the nasal spray vaccine, a blocked or very runny nose is about the only AR observed. 

All of these have been widely described, yet there is less information as to how repeated immunization relates to ARs or how ARs relate to higher risk of attaining Influenza infection.

This survey reveals information as to the factors related to ARs in influenza vaccination of HCWs in several hospital settings in Japan.  

                   Here are my considerations for corrections and changes.

Note: The naming of the vaccine should be the same throughout the entire paper, either tetravalent or quadrivalent vaccine , either one is correct but stick to one of them.

Response: Thank you for the comments. We have consistently used “quadrivalent” in the revised manuscript.

Abstract

Conclusion: Those who had ARs, notably localized ones, were  significantly more likely to incur influenza infection. Individuals who have had ARs to influenza  vaccination should be more cautious about seasonal influenza illness

I would be more concise as to being more cautious, for example: strictly adhering to Non-pharmaceutical interventions in the hospital setting as well as in the community setting and household.

Response: Thank you for the valuable comments. As per your advice, we revised sentence as follows:

“Individuals reporting ARs to influenza vaccination should strictly adopt non-pharmaceutical preventive measures in the hospital and community settings and at home.” (Lines 29–31)

Background

Lines 51-52  Therefore, prevention of infection is and will continue to be important for this disease, which requires careful attention to determine infection trends. Change: careful attention to close surveillance to monitor infection trends

Response: Thank you for the valuable comments. We have revised the sentence as follows:

“Therefore, prevention of infection by close surveillance to monitor infection trends is and will continue to be important for this disease.” (Lines 53–54)

Lines 53-56  Influenza vaccines are effective in preventing infection in healthy adults; however,   effectiveness is reported to be lower among the elderly than among the youth [8]. Despite   this, the influenza vaccine has been reported to be effective in preventing severity and   hospitalization in addition to preventing infection [9] and may be important for the elderly.  

Rephrase: Influenza vaccines are effective in preventing infection in healthy adults; however,   effectiveness is reported to be lower among the elderly [8]. Despite this, influenza vaccine has been reported to be effective in preventing severity and   hospitalization [9] especially among population at risk of severe complications such as  the elderly and those with comorbidities at any age.

Response: Thank you for the valuable comments. We have rephrased the text according to your advice as follows:

“Influenza vaccines are effective in preventing infection in healthy adults; however, their efficacy is reported to be lower among the elderly [8]. Despite this, influenza vaccine has been reported to be effective in preventing severity and hospitalization [9], particularly among patients at risk of severe complications, such as the elderly, and those with comorbidities at any age.” (Lines 55–59)

Lines 62-64: There is limited literature on adverse reactions post influenza vaccination, of which, adverse reactions have been reported to more likely occur in women than in men with both the quadrivalent and trivalent vaccines [12] and that adverse reactions are less likely to occur in older people [11].

The structure of the sentences is awkward. Change to: There is limited literature on adverse reactions post influenza vaccination, most of which refers to adverse reactions being more likely to occur in women than in men with both the quadrivalent and trivalent vaccines [12] and that adverse reactions are less likely to occur in older people [11].

Response: Thank you for the valuable comments. We changed the text according to your advice as follows:

“Although the literature on adverse reactions post influenza vaccination is limited, adverse reactions are more likely to occur in women than in men with both the quadrivalent and trivalent vaccines [12] and less likely to occur in older people [11].” (Lines 64-67)

2.Materials and Methods

An explanation as to what is considered history of influenza vaccination is needed. Did you considere any previous immunization or the last season, last 2-3-4-5 seasons?

Response: Thank you for the valuable comments. We did not investigate whether past influenza vaccinations were trivalent or quadrivalent. However, given that the participants received the influenza vaccine from the hospital and at an almost yearly interval, as hospital personnel, we believe that they have been vaccinated almost exclusively with quadrivalent vaccinations since the 2015-2016 season. This is noted in the study limitations. (Lines 280–283)

The questionnaire distributed to the participants stated that it is to be used for research purposes, and informed consent was obtained in writing for the publication of 76 the study results. written informed consent was obtained for the publication of the study results

Response: Thank you for the valuable comments. We have revised the manuscript according to your advice as follows:

“The questionnaire distributed to the participants stated that data collected are to be used for research purposes, and written informed consent was obtained for the publication of the study results. ” (Lines 77–80)

Setting Line 82 The study questionnaire and immunization questionnaire were distributed to the  participants ? There are two questionnaires ? what is the difference between them ?

 Response: Thank you for the valuable comments. To address your concerns, we have added the following text to the “Questionnaire Items” section.

“We distributed two types of questionnaires. The first questionnaire collected participants' demographic data and identified participants who had received the vaccination. The second questionnaire collected data regarding adverse reactions to the vaccine and was collected 10 days after immunization.” (Lines 89–92)

Conclusions : Same comment as in abstract conclusions

Response: Thank you for the valuable comments. Following your advice, we revised the sentence as follows:

“Individuals reporting adverse reactions to influenza vaccination should strictly adhere to non-pharmaceutical preventive measures in the hospital and community settings and at home.” (Lines 296–298)

Reviewer 2 Report

1. those who had adverse reactions, especially local adverse reactions, were significantly more likely to be infected with influenza. Individuals who have had adverse reactions to influenza vaccination should be more cautious about seasonal influenza illness. 

According to this conclusion, the elderly over the age of 65 after vaccinated with low incidence of adverse reactions, the possibility of contracting the influenza will be low, then the urgency of the  influenza vaccine in the elderly than young people is reduced, the conclusion and the elderly annual influenza vaccination recommendations contradict each other, and effectiveness of the vaccine in healthy adults is 70%-90%,But the finding that only 17% to 53% of the elderly are also contradictory.

Therefore, please conduct in-depth and scientific discussion, rationalize the relationship between adverse reactions of immunization vaccine, immunization effect of vaccine and influenza reinfection, and draw the correct conclusion.

2. 64 Is the type of influenza infected in the person consistent with the serotype of influenza virus contained in the vaccine? If so, people who do not have a strong immune response (adverse reactions) are likely to have influenza again.Is there a significant difference in the proportion of 64 people infected with influenza in each age group?

3. In line 186-187, is the proportion of male and female in the total number correct? Why the sum of the two is greater than 100%?

Author Response

Reviewer 2’s Comments and Suggestions for Authors:

  1. those who had adverse reactions, especially local adverse reactions, were significantly more likely to be infected with influenza. Individuals who have had adverse reactions to influenza vaccination should be more cautious about seasonal influenza illness. 

According to this conclusion, the elderly over the age of 65 after vaccinated with low incidence of adverse reactions, the possibility of contracting the influenza will be low, then the urgency of the  influenza vaccine in the elderly than young people is reduced, the conclusion and the elderly annual influenza vaccination recommendations contradict each other, and effectiveness of the vaccine in healthy adults is 70%-90%,But the finding that only 17% to 53% of the elderly are also contradictory.

Therefore, please conduct in-depth and scientific discussion, rationalize the relationship between adverse reactions of immunization vaccine, immunization effect of vaccine and influenza reinfection, and draw the correct conclusion.

Response: Thank you for the valuable comments. Elderly participants referred to in the Background were in their late 70s to over 80 years of age, as indicated by the median age of 78 years. In contrast, we included participants aged <70 years, with the oldest age group being between 65 and 70 years of age. Thus, the discrepancy can be attributed to the different age groups. The fact that the participants in this study were aged <70 years was emphasized in the Results and in section “4.1 Incidence of Influenza and Adverse Reactions.” (Lines 217-223)

 We added the following facts to the Results section. (Lines 142–144)

“All participants were aged between 20 and 70 years; 622 (41.66%) were aged <40 years, 818 (54.79%) were aged 40–64 years, and 53 (3.55%) were aged ≥65 years.”

In section 4.2 Age and Adverse Reactions, “Vaccine efficacy” was revised to “Vaccine efficacy in preventing the onset of influenza. (Line 231)

In addition, we added the following: The influenza vaccine is less effective in preventing disease in the elderly than in healthy adults [8]. However, it is effective not only in preventing the onset of influenza but also in preventing hospitalization (prevention of severe illness) [8], which may play an important role in the elderly, who are frequently hospitalized.

In the elderly, vaccination is expected to prevent the onset and severity of disease, whereas in healthy adults, vaccination is effective for preventing the onset of disease in the vaccinee and transmission of infection to the elderly or immunocompromised individuals [9]. In particular, healthcare workers have many opportunities to come into contact with patients with underlying diseases, and vaccination is meaningful not only for preventing their own infection, but also from the perspective of protecting patients. (Lines 238–247)

Further, we added the following to the "Limitations”:The participants in this study were hospital employees in good health aged between 20 and 70 years. Therefore, people in their late 70s and aged >80 years, who have high rates of serious illness and mortality, were not included in the study. (Lines 284–287)

Finally, we have revised the Conclusions as follows:

In this study, adverse reactions were less likely to occur with increasing age in healthcare professionals, especially in participants aged >65 years, which is similar to the findings of previous studies. Moreover, men were significantly less likely to have an adverse reaction than women.

Furthermore, participants with a history of influenza vaccination were more likely to have adverse reactions. In terms of the relationship between adverse reactions and influenza morbidity, participants who had adverse reactions, especially local adverse reactions, were significantly more likely to contract influenza.

Individuals reporting adverse reactions to influenza vaccination should strictly adhere to non-pharmaceutical interventions in the hospital and community settings and at home. (Lines 288–298)

  1. 64 Is the type of influenza infected in the person consistent with the serotype of influenza virus contained in the vaccine? If so, people who do not have a strong immune response (adverse reactions) are likely to have influenza again.Is there a significant difference in the proportion of 64 people infected with influenza in each age group?

Response: Thank you for the valuable comments. We have added the influenza subtypes included in the influenza vaccine in section "2.2. Participants and Setting" as follows:

“The influenza vaccine administered in the 2019-2020 season included the following four subtypes: A/Brisbane/02/2018(H1N1)pdm09-like virus, A/Kansas/14/2017 (H3N2)- like virus, B/Colorado/06/2017- like virus (B/Victoria/2/87 lineage), B/Phuket/3073/2013- like virus (B/Yamagata/16/88 lineage).” (Lines 83–86)

Further, we have added the following in the Results section:

“Although data on individual influenza subtypes in the 64 participants who contracted influenza were not available, the influenza types and subtypes that were prevalent in Japan during the 2019-2020 season were A/H1N1pdm09 (85%), A/H3N2 (2%), and B (12%; Victoria strain 98.7%, Yamagata strain 0.6%, and unknown strain 0.7%).”

In addition, influenza vaccines administered in 2019 contained the following strains: A/Brisbane/02/2018(H1N1)pdm09-like virus, A/Kansas/14/2017 (H3N2)- like virus, B/Colorado/06/2017- like virus (B/Victoria/2/87 lineage), and B/Phuket/3073/2013- like virus (B/Yamagata/16/88 lineage) [3]. Therefore, it can be considered that the strains in the vaccine and the affected participants were almost identical. (Lines 148–157)

Table 1 presents the incidence of influenza in the current study. A small proportion of participants (3.55%; 53/1493), were aged ≥65 years. Among participants aged <40 years and 40–60 years 33 of 622 (5.31%) and 31 of 818 (3.79%) contracted influenza, respectively. However, there were no cases (0/53; 0%) among those aged ≥65 years. We have added this to section “4.1 Incidence of Influenza and Adverse Reactions” as follows:

“In our study, no patient aged between 65 and 70 years contracted influenza, which was the lowest incidence for this age group in Japan.” (Lines 217–223)

  1. In line 186-187, is the proportion of male and female in the total number correct? Why the sum of the two is greater than 100%?

Response: Thank you for the valuable comments. It means "the incidence of adverse reactions was 82.4% in men and 91.8% in women. We rephrased the text as follows:

“In a Japanese study of 3,275 cases at 97 facilities, the incidence of adverse reactions was 86.8%. The incidence of adverse reactions was higher in women than in men (91.8% vs. 82.4%), [11,12].” (Lines 202–204)

Reviewer 3 Report

In the current study, adverse reactions (ARs) were investigated among 1,493 hospital employees who received the 2019 influenza vaccination. It was shown that ARs were less common among men than among women, and were less likely to occur with increasing age, especially among people over the age of 65. It was confirmed that participants with a history of influenza vaccination were more likely to have ARs. Interestingly, individuals who had ARs, notably local ones, were significantly more likely to incur influenza infection, suggesting that individuals with ARs should be more cautious about seasonal influenza. The major concern the reviewer has is how reliable the last result is. In this study, only 64 out of the 1,524 participants got the flu (line 197). Furthermore, ARs after vaccination are generally most often associated with exaggerated immune responses induced by vaccination. It sounds strange to me that vaccine recipients with higher immune responses are more susceptible to influenza. As shown in Lines 242-243, the authors simply wrote they did not know the mechanism of it, and did not give any consideration as to why it happened.

Minor points

1.     Line 211: The obsolete word, “suppressor T cells” should be replaced by “regulatory T cells”.

2.     Lines 245-246: I do not understand the sentence “the absence of adverse reactions in influenza vaccines does not necessary imply the absence of antibodies”. Fundamentally, flu-vaccine recipients with or without adverse reactions produce antibodies against influenza virus.

Author Response

Reviewer 3’s Comments and Suggestions for Authors:

In the current study, adverse reactions (ARs) were investigated among 1,493 hospital employees who received the 2019 influenza vaccination. It was shown that ARs were less common among men than among women, and were less likely to occur with increasing age, especially among people over the age of 65. It was confirmed that participants with a history of influenza vaccination were more likely to have ARs. Interestingly, individuals who had ARs, notably local ones, were significantly more likely to incur influenza infection, suggesting that individuals with ARs should be more cautious about seasonal influenza. The major concern the reviewer has is how reliable the last result is. In this study, only 64 out of the 1,524 participants got the flu (line 197). Furthermore, ARs after vaccination are generally most often associated with exaggerated immune responses induced by vaccination. It sounds strange to me that vaccine recipients with higher immune responses are more susceptible to influenza. As shown in Lines 242-243, the authors simply wrote they did not know the mechanism of it, and did not give any consideration as to why it happened.

Response: Thank you for the valuable comments. To address this concern, we have added the following text to section 4.1 “Incidence of Influenza and Adverse Reactions”

The incidence of influenza in Japan during the same observation period, was 5.7% [3]. Considering that the participants in this study were healthcare workers, who are more knowledgeable regarding infection control measures than the general public, we believe that the incidence rate is reasonable. (Lines 212–216)

Further, we added the following to section 4.5 “Influenza Morbidity and Adverse Reactions”:

“In this study, the adjusted ORs suggested a tolerable incidence of influenza morbidity. With regard to the severe acute respiratory syndrome coronavirus 2, greater adverse reactions experienced post-vaccination translates to a higher antibody production [17], which reduces the susceptibility to coronavirus disease 2019. In this study, more adverse reactions resulted in a higher incidence of influenza. The mechanism for this is unknown and has not been reported in the existing literature. In the current study, a history of influenza vaccination significantly increased the incidence of any adverse reaction (either local or systemic) and local adverse reactions. However, 95.08% of participants had a history of previous influenza vaccination. This may explain why the occurrence of adverse reactions did not necessarily reduce the incidence of influenza.” (Lines 264–274)

Minor points

  • Line 211: The obsolete word, “suppressor T cells” should be replaced by “regulatory T cells”.

Response: Thank you for the valuable comment. We revised “suppressor T cells” to “regulatory T cells”. (Line 233)

  • Lines 245-246: I do not understand the sentence “the absence of adverse reactions in influenza vaccines does not necessary imply the absence of antibodies”. Fundamentally, flu-vaccine recipients with or without adverse reactions produce antibodies against influenza virus.

Response: Thank you for the valuable comment. We have deleted the sentence.

Round 2

Reviewer 3 Report

I have no serious criticisms.